# Melatonin Promotes the Development of Secondary Hair Follicles in Adult Cashmere Goats by Activating the *Keap1-Nrf2* Signaling Pathway and Inhibiting the Inflammatory Transcription Factors *NFκB* and *AP-1*

**DOI:** 10.3390/ijms24043403

**Published:** 2023-02-08

**Authors:** Xiaogao Diao, Chunhui Duan, Lingyun Yao, Jiaxin Qin, Liwen He, Wei Zhang

**Affiliations:** 1Department of Animal Nutrition and Feed Science, College of Animal Science and Technology, China Agricultural University, No. 2, Yuan Ming Yuan West Road, Beijing 100193, China; 2College of Animal Science and Technology, Agricultural University of Hebei, Baoding 071000, China

**Keywords:** melatonin, secondary hair follicles, cashmere, *Keap1*, *Nrf2*, *NFκB*, *AP-1*

## Abstract

Exogenous melatonin (MT) has been used to promote the growth of secondary hair follicles and improve cashmere fiber quality, but the specific cellular-level mechanisms involved are unclear. This study was carried out to investigate the effect of MT on the development of secondary hair follicles and on cashmere fiber quality in cashmere goats. The results showed that MT improved secondary follicle numbers and function as well as enhanced cashmere fiber quality and yield. The MT-treated goat groups had high secondary-to-primary ratios (S:P) for hair follicles, greater in the elderly group (*p* < 0.05). Antioxidant capacities of secondary hair follicles improved fiber quality and yield in comparison with control groups (*p* < 0.05/0.01). Levels of reactive oxygen and nitrogen species (*ROS, RNS*) and malondialdehyde (*MDA*) were lowered (*p* < 0.05/0.01) by MT. There was significant upregulation of antioxidant genes (for *SOD-3; GPX-1; NFE2L2*) and the protein of nuclear factor (*Nrf2*), and downregulation of the *Keap1 protein*. There were significant differences in the expression of genes for secretory senescence-associated phenotype (SASP) cytokines (*IL-1β, IL-6, MMP-9, MMP-27, CCL-21, CXCL-12, CXCL-14, TIMP-1,2,3*) plus their protein of key transcription factors, nuclear factor kappa B (*NFκB*) and activator protein-1 (*AP-1*), in comparison with the controls. We concluded that MT could enhance antioxidant capacity and reduce *ROS* and *RNS* levels of secondary hair follicles through the *Keap1-Nrf2* pathway in adult cashmere goats. Furthermore, MT reduced the expression of the SASP cytokines genes by inhibiting the protein of *NFκB* and *AP-1* in the secondary hair follicles in older cashmere goats, thus delaying skin aging, improving follicle survival, and increasing the number of secondary hair follicles. Collectively, these effects of exogenous MT enhanced the quality and yield of cashmere fibers, especially at 5–7 years old.

## 1. Introduction

China has the world’s largest population of cashmere goats, with a stock of more than 70 million animals, and cashmere fiber is the most valuable commodity they produce. In 2021, China’s total cashmere production reached 17,800 tons, accounting for more than 70% of the world’s total production, and more than 60% of China’s production is exported, accounting for about 70% of the international market. In recent years, the international market demand for high-grade cashmere has exceeded the supply [1,2]. There is also a trend of increasing improvement in the quality of the cashmere product.

Inner Mongolia cashmere goats are internationally renowned for their excellent cashmere fiber quality [3]. The annual cashmere yield of each goat is 600-800 g with a fiber diameter of 11–16 µm, and the growth of cashmere in these goats is highly seasonal [4,5]. Cashmere traits are affected by both non-genetic factors and genetic factors. Non-genetic factors, including sex, age, production year, age of dam, type of birth, herd management, etc. [6,7]. FGF12, SEMA3D, EVPL, SOX5, and other genes related to cashmere traits were screened out in the latest studies, and the overlapping genes Wnt10a and CSN3 associated with cashmere traits were found in the comparative study with Liaoning cashmere goat [8,9,10]. Cashmere grows mainly in August each year and is harvested by combing out at the end of April in the following year. The period from May to August is regarded as the ‘non-growing cashmere period’, although it is during this period that the development of secondary hair follicles occurs annually in adult goats. Studies have shown that secondary hair follicle growth is associated with the secretion of melatonin (MT) from the pineal gland [11,12,13]. MT is involved in the regulation of seasonal rhythms as well as having anti-aging, skin protective, anti-oxidative, and anti-tumor properties [14,15,16,17]. MT has been administered to fur-bearing animals such as mink, blue fox, and rex rabbit to promote hair follicle activity [18,19,20,21]. Our laboratory has conducted studies on goats with implantation of MT during the non-growing cashmere period showing that MT can promote the growth of secondary hair follicles, increase their numbers, and improve the quality of cashmere fibers [2,22]. However, the nature of the molecular mechanisms by which MT exerts its effects on these hair follicles is still unclear. Most of the previous studies with MT have used young cashmere goats (2 or 3 years old), and there is little information about its effects on older animals, yet the productive life of cashmere goats is about 6 to 7 years. With the increasing age of the goats, the yield decreases, and cashmere fibers become shorter and thicker, leading to a reduction in quality [23,24,25]. It is believed that factors such as oxidative stress, accumulation of reactive oxygen and nitrogen species (*ROS, RNS*), skin aging, and deterioration of the microenvironment of the hair follicles are responsible for the decline in the number of hair follicles [26,27], and the secretion of MT also declines with age [28,29]. So, we hypothesized that the provision of exogenous MT to older cashmere goats might preserve the activity of secondary hair follicles through the enhancement of antioxidant and anti-aging processes in the skin.

This study was conducted to investigate the molecular mechanisms through which MT exerts its effects on the development and functions of secondary hair follicles of cashmere goats at different ages. Its findings would provide an improved theoretical basis for the use of exogenous MT within the cashmere goat industry.

## 2. Results

### 2.1. Cashmere Quality and Yield

Generally, the treatment of goats with MT improved both the quality and yield of cashmere. The mean cashmere fiber diameters of goats in the MT groups ranged between 14.02 and 14.34 µm and were lower (*p* < 0.05/0.01) than those of the CK groups, which ranged between 14.65 and 15.49 µm (Figure 1A). The mean lengths of the fibers were longer (*p* < 0.01 in most cases) in MT goats (ranging from 8.25 to 10.48 cm) than in controls (ranging from 6.99 to 8.03 cm). (Figure 1B). The mean yields of cashmere, whether gross yields or adjusted for the live weight of the animals, were almost always higher (*p* < 0.01/0.05) in the MT groups (Figure 1C,D).

### 2.2. Primary and Secondary Hair Follicles

Results for primary hair follicle density (PFD), secondary hair follicle density (SFD), and the ratio of secondary to primary hair follicles (S:P) are presented in Figure 2. The preponderance of secondary hair follicles in skin sections taken from goats in the MT groups, compared with controls, is visible in the photomicrographs (Figure 2A,B). There was no effect of MT on PFD (Figure 2C). However, in contrast, the means for SFD and S:P in most cases were higher (*p* < 0.01 and *p* < 0.05) in the MT groups (Figure 2D).

### 2.3. Antioxidant Capacity of Serum and Skin

In some cases, MT groups had higher (*p* < 0.05) mean values for *CAT, T-AOC, SOD,* and *GSHPx* than the CK groups (serum-Figure 3A–D; skin-Figure 4A–D) and lower (*p* < 0.01/0.05) mean values for *MDA* (serum-Figure 3E; skin-Figure 4E).

### 2.4. Markers of Oxidative Stress

Generally, the mean levels for markers of oxidative stress (*ROS* and *RNS*) were lower (*p* < 0.05/0.01) for serum (Figure 5A,B) and skin (Figure 6C,D) of goats in the MT groups in comparison with their controls (CK groups).

### 2.5. Differential mRNA Expression between MT vs. CK

The differential expression genes (DEGs) analysis showed that there were 3051 DEGs expressed in the goats at 2 years of age: 1658 were upregulated and 1393 were downregulated. At 7 years of age, there were 148 DEGs, of which 30 were upregulated and 118 were downregulated. The volcano plots and heat maps show the DEGs of each group (Figure 6A–D). At 2 years of age, A*TP-8/6, COLA-1, COLA-2, KRTAP8-1, KRT-1/10, HSP-90AA1*, and *CCL-26* were highly expressed, as were *KRTAP1-1, PSORS1C-2, KRTAP-8-1, KRTAP-24-1*, and *MMP-9* at 7 years of age. We also recorded DEGs related to oxidative stress (*GPx-1, PTGS-1, GSTO-2, SOD-3, APOE, DUOX-1,* and *NFE2L2*) and performed GO and KEGG analyses for all DEGs. At 2 and 7 years of age, most differential genes from GO analysis were related to cellular organizational processes (Figure 7A,B). KEGG analysis (Figure 7C) showed that DEGs at 2 years of age were enriched in the phosphatidylinositol 3-kinase-protein kinase B (*PI3K-AKT*), interleukin-17 (*IL-17*), mitogen-activated protein kinase (*MAPK*), nuclear factor kappa B (*NFκB*) and other classical signaling pathways and at 7 years of age, arachidonic acid metabolism, sphingolipid metabolism, and the p53 and NFκB signaling pathways were enriched (Figure 7D).

### 2.6. Antioxidant Gene Expression and Protein Levels

Levels of cyclooxygenases (*COX-1* and *COX-3*), superoxide dismutase-3 (*SOD-3*), glutathione peroxidase-1 (*GPx-1*), and nuclear factor erythroid 2-related factor 2 (*NFE2L2*) from the RNAseq data were generally higher in the skin and secondary hair follicles of goats in the MT groups in comparison with controls (CK groups) at 2 and 7 years of age (Figure 8A). Furthermore, q-PCR showed that expression of these genes was almost always significantly (*p* < 0.01) higher in the MT groups at these ages (Figure 8A). Western blots showed lower levels of *Keap1* and higher levels of *Nrf2* proteins for the skin and secondary hair follicles of goats in the MT groups in comparison with the controls at these ages (Figure 8B).

### 2.7. Anti-Aging Gene Expression and Protein Levels

Analysis (q-PCR) of the secretory senescence-associated phenotype (SASP) genes in the skin and secondary hair follicles of 7-year-old cashmere goats showed lowered expression (*p* < 0.01/0.05) in the MT group in comparison with controls for genes encoding interleukins (*IL-1β, IL-6*) and matrix metalloproteinases (*MMP-9, MMP-27*) and chemokine ligand genes (*CCL-21, CXCL-12, CXCL-14*), and elevated expression (*p* < 0.01/0.05) of genes for tissue inhibitor of metalloproteinases (*TIMP-1, TIMP-2, TIMP-3*) (Figure 9A). At the same age, MT-treated goats had lower levels (Western blotting) of the nuclear transcription factors *C-jun, C-fos,* and *P65* than controls (Figure 9B).

## 3. Discussion

These results show that the treatment-related effects of exogenous MT on improving the quality of fiber produced by adult cashmere goats tend to diverge further from their controls with the increasing age of the goats. These include increased S:P ratio of hair follicles and changes in expression of secondary hair follicle-related genes, including enrichment of multiple pathways related to hair follicle development. Moreover, the present study shows that treatment with MT improves the antioxidant capacity and reduces markers of oxidative stress in circulating blood and the skin by regulating the *Keap1-Nrf2* pathway. In addition, treatment with MT improved the anti-aging ability of these follicles by reducing the expression of the transcription factors, *NFκB* and activator protein-1 (*AP-1*), inhibiting the SASP genes (for *IL-1β, IL-6, MMP-9, MMP-27, CXCL-12, CXCL-14, CCL-21*) and upregulating gene expression for *TIMP-1, TIMP-2,* and *TIMP-3*, thus delaying skin senescence and improving the microenvironment of the secondary hair follicles. The ultimate effect of these changes in gene expression is an increase in the number of active secondary hair follicles with consequent improvement of the quality of cashmere fibers.

Cashmere is the main product of cashmere goats, and it is not only different from wool in fiber structure but also has a much higher economic value than wool. The quality of the cashmere fleece is determined by the length and diameter of its individual fibers, and these are affected by factors such as sex, growth stage, nutrition, and endocrine status [30,31,32,33]. The cashmere quality at 1 year old is the highest among all age groups, and with the increase in age, the cashmere quality had a decreasing trend. Some reports show that low energy and protein levels can improve the fineness and length of cashmere, trace elements can improve the strength of cashmere fiber, and the change in blood hormone levels will affect the character of cashmere [34,35]. The quality is highest in 1/2-year-old goats and thereafter declines with the increasing age of the animals [36,37]. In the present study, treatment with MT caused an increase (above that of controls) in fiber quality at 5 and 6 years of age, indicating that MT has counteracted this trend of reduced fiber quality with the aging of the goats. Our previous studies have shown that provision of young goats with exogenous MT at the onset of the fiber growth stage (April to August) could increase the number of active secondary hair follicles [6,38]. The present findings extended this by showing that MT increased secondary hair follicle density and the S:P ratio of goats at 5, 6, and 7 years of age, indicating that MT maintains its effects at these ages.

Ultraviolet (UV) radiation can accelerate skin aging and cause damage to hair follicles [39,40]. Aging leads to reduced ability of the skin to scavenge free radicals, which fosters the deposition of *ROS* in the skin, leading to a poorer microenvironment for hair follicles [41,42,43,44]. Hair follicle stem cells dominate hair follicle morphogenesis and the periodic growth of fiber, but their ability to self-renew is affected by aging [45,46,47]. The present results show that treatment with MT raises antioxidant indices, thus improving the antioxidant capacity of the secondary hair follicle, as it does in mice [48,49], and raising the number of secondary hair follicles.

Hair follicles are the derived organs of skin in cashmere goats and are divided into the primary hair follicles and the secondary hair follicles [50]. The primary hair follicle grows coarse hair, and the secondary hair follicle grows cashmere. The growth of cashmere has apparent periodicity, expands on the body surface in August each year, and falls off naturally in April the following year. Cashmere growth and shedding are determined by the cycle of secondary hair follicles, which includes periods of anagen, catagen, and telogen [51,52]. Hair follicle morphogenesis is dependent on the interaction between epidermal and dermal cells, which requires the expression of genes coding for proteins that are involved in signaling pathways, cell migration, and cell aggregation. Wnt, bone morphogenetic protein (*BMP*), and fibroblast growth factor (*FGF*) signaling pathways have been shown to play critical roles in hair follicle development, and proteins such as keratin (*KRT*) and collagen 1a (*COLA*) are involved in hair follicle cycle changes [53,54]. Study shows that the secondary hair follicles exposed to MT show differential changes in genes coding for kidney androgen-regulated protein (*KAP)*, *KRT, COLA, BMP-2,* and *BMP-4* [55]. These genes are all involved in hair follicle development. Cyclooxygenases are the rate-limiting enzymes that produce prostaglandins, and two of these, *COX-1* and *COX-2*, have an integral role in keratinocyte differentiation [56,57]. The *COX-1* gene is expressed in skin and hair follicles (mice and humans) and has a central role in skin repair [58], and *COX-1* expression may be upregulated by flavonoids to exert their skin-protective activities [59]. So, it is possible that MT achieved its protective effects on the hair follicles of the goats in this study likewise by upregulating *COX-1*.

ROS and RNS are by-products of cell metabolism [60,61]. Accumulation of high concentrations of these compounds within cells can lead to oxidative damage and cause functional degradation of tissues. Various environmental factors, including UV radiation and atmospheric pollutants (e.g., ozone, particulate matter), can increase the production of *ROS* and *RNS* in the skin, causing damage to the skin and hair follicles [62,63,64]. Oxidative stress may be a key factor leading to hair aging or alopecia [65]. The transcription factor *Nrf2* is widely expressed in skin and hair follicles, and its activation prevents peroxide-induced inhibition of hair follicle growth [66,67]. Substances with antioxidant properties operate by activating Nrf2 expression [68,69,70]. Tripathi et al. report that MT can reduce oxidative stress by scavenging free radicals directly, but it can also achieve this indirectly by activating the Kelch-like ECH-associated protein-1-*Nrf2*-antioxidant responsive element (*Keap1-Nrf2-ARE*, in short-*Keap1-Nrf2*) pathway to improve the body’s antioxidant capacity [71]. The present results in goats show that MT has activated the *Keap1-Nrf2* pathway to improve antioxidant capacity and reduce the oxidative stress of secondary hair follicles, as well as promote their proliferation.

Skin aging is accompanied by decreased blood and nutrient supply to hair follicles, which leads to fibrosis of the dermis, impaired follicle remodeling, and reduced production of hair fibers [72,73]. *SASP* is a general term for cytokines secreted by senescent cells. They can induce inflammation and accelerate cell senescence through *ROS*-mediated pathways [74]. The *SASP* family includes interleukins, chemokines, growth factors, matrix metalloproteinases (*MMPs*), and tissue inhibitors of metalloproteinases. Genes for *MMPs* are expressed in hair follicles and other skin tissues, and the functions of *MMPs* are disrupted by skin diseases and damage from UV radiation. Upregulation of *MMP-9* is reported to be a major factor for hair follicle degeneration in alopecia [75,76,77]. Moreover, exogenous MT has been shown to counteract the adverse effects of UVB radiation by reducing MMPs (1 and 3), increasing TIMP levels by inhibiting the c-Jun N-terminal kinase-AP-1 (JNK-AP-1) pathway, and delaying senescence by down-regulating NFκB and inhibiting the SASP cytokines [78,79]. In the case of chemokine ligand genes, *CXCL-12* is increased in burned skin, and blockade of the *CXCR-4-CXCL-12* interaction promotes the recovery of burned skin [80]. In addition, expression of *CXCL-9* and *CXCL-10* is elevated in mice with alopecia areata lesions [81]. Many of the alterations of gene expression that are protective to hair follicles were enacted in response to MT treatment of goats in the present study and led us to infer that MT has delayed senescence in secondary hair follicles by reducing expression of *NFκB* and *AP-1*, thus lowering the production of the SASP cytokines.

## 4. Materials and Methods

### 4.1. Animals and Management

Figure 10 shows our workflow of this experimental design. Adult superfine cashmere goats within each of the 6 age groups (2, 3, 4, 5, 6, 7 years) were randomly allocated to melatonin-treated (MT) or untreated control (CK) groups (*n* = 15). Goats in the MT groups were implanted s.c. with MT pellets on 2 occasions (late April and late June) to provide a dose of 2 mg/kg each time. (The sustained release period of these pellets is about 60 days (Kangtai Biotechnology Co., Ltd., Beijing, China)). The goats were grazed on pasture (8 h per day) and provided with supplementary feed (the supplementary concentrate consisted of 70% corn and 30% concentrate feed and was purchased from a local feed company (Baotou Jiuzhoudadi Biotech Company, Baotou, China) in accordance with the feeding management system of the research facility. Water was freely available.

### 4.2. Samples Collection and Processing

Skin and blood samples were collected from each animal 30 days after the date of implantation with MT. Four pieces of skin (1 cm^2^) were obtained from the left shoulder blade by the skin sampler (HeadBio, Beijing, China) under painless handling. One of these was fixed with 4% paraformaldehyde for subsequent microscopic (Leika, Wetzlar, Germany) examination, and one was stored in liquid nitrogen for PCR and Western blot determinations. The other 2 pieces were used for reactive oxygen species (*ROS*) and reactive nitrogen species (*RNS*) analyses. A single sample of blood (5 mL) from each animal was collected by jugular venipuncture between 2200 and 2300 h, allowed to coagulate, and the resultant serum was stored in liquid nitrogen for later determination of serum antioxidant indices. Each animal’s total annual cashmere output was collected by combing out the fleece the following year and weighing it to determine yield, and fiber length and diameter were measured according to the procedures described by Yang [23]. Antioxidant indices (catalase (*CAT*), total antioxidant capacity (*T-AOC*), superoxide dismutase (*SOD*), glutathione peroxidase (*GSHPx*), malondialdehyde (*MDA*)) and *ROS* of skin and serum were measured using a kit purchased from Nanjing Jiancheng Bioengineering and following the supplier’s instructions. *RNS* levels were determined using a goat-specific ELISA kit from *Shanghai mlbio* (Shaihai, China).

### 4.3. mRNA Transcriptome Analysis

The trizol method was used to extract total RNA from skin tissues, and agarose gel electrophoresis was used to ensure the integrity of mRNA. The mRNA samples were sent to Majorbio (Shanghai, China) for detection and library construction. The software (DESeq2 and Goatools) provided by the Majorbio cloud platform (www.majorbio.com, accessed on 16 December 2022) was used to analyze mRNA expression and the differential expression of mRNA (DEmRNA) using gene ontology (GO) and the Kyoto Encyclopedia of Genes and Genomes (KEGG).

### 4.4. RT-qPCR and Western Blotting

A total of 100 mg of skin was ground to extract total RNA, and reverse transcription into cDNA was conducted according to RT-qPCR steps and processed according to formula 2^−ΔΔCT^. Table 1 shows the primer sequences. Measurement of skin protein used the bicinchoninic acid (BCA) method to quantify the extracted protein and adjust all samples to the same concentration. Electrophoresis was performed on a 10% SDS-PAGE gel and then transferred to a polyvinylidene difluoride (PVDF) membrane. The PVDF membrane was blocked with 5% skimmed cow’s milk, followed by rabbit anti-*Nrf2* (1:2000), goat anti-*Keap1* (1:2000), rabbit anti-*P65* (1:2000), rabbit anti-*C-fos* (1:1000), rabbit anti-*C-Jun* (1:1000), rabbit anti-*β-actin* (1:6000) (Servicebio, Wuhan, China) overnight at 4 °C, then incubated with secondary antibodies. Peroxidase activity on the membrane was visualized on X-ray film using an enhanced chemiluminescence (GE Bioscience, Newark, NJ, USA) Western blotting detection system.

### 4.5. Statistical Analyses

Data were collated and organized on an Excel spreadsheet. The SPSS28 (SPSS Inc., Chicago, IL, USA) general linear model was used to compare the hair follicle indices, cashmere yield, and fiber diameter and length in each year cohort. Results were expressed as the mean ± SD, and the differences between means were considered significant at *p* < 0.05.

## 5. Conclusions

In this study, we found that treatment of goats at various ages with exogenous MT could affect the expression of secondary hair follicle-related antioxidant and anti-aging genes in a manner that improves follicular activity. These changes in gene expression reduce oxidative stress and appear to be manifested by MT-stimulated regulation of the *Keap1-Nrf2* pathway. Moreover, the resultant intracellular effects include a reduction in the transcription factors *NFκB* and *AP-1* and lowered levels of the *SASP* cytokines. These changes delay skin aging processes and improve the microenvironment of secondary hair follicles, ultimately increasing the number of secondary hair follicles and improving the quality and yield of cashmere fibers (Figure 11). Collectively, exogenous MT enhanced the quality and yield of cashmere fibers and was more effective in older cashmere goats (>5 years old).

## Figures and Tables

**Figure 1 ijms-24-03403-f001:**
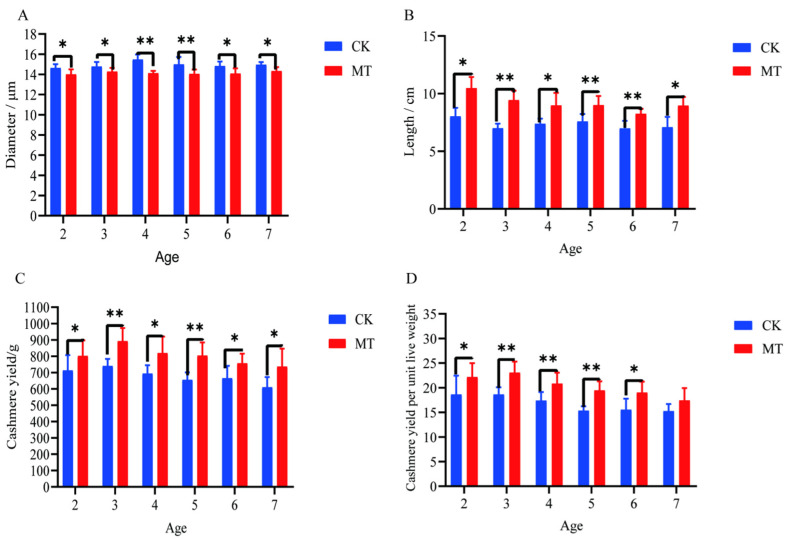
Cashmere fiber indices of cashmere goats treated with melatonin (MT) or not treated (CK) at different ages. (**A**) Fiber diameter; (**B**) fiber length; (**C**) cashmere yield; (**D**) cashmere yield per unit live weight. All the values are the mean ± standard deviation. * *p* < 0.05, ** *p* < 0.01, CK vs. MT.

**Figure 2 ijms-24-03403-f002:**
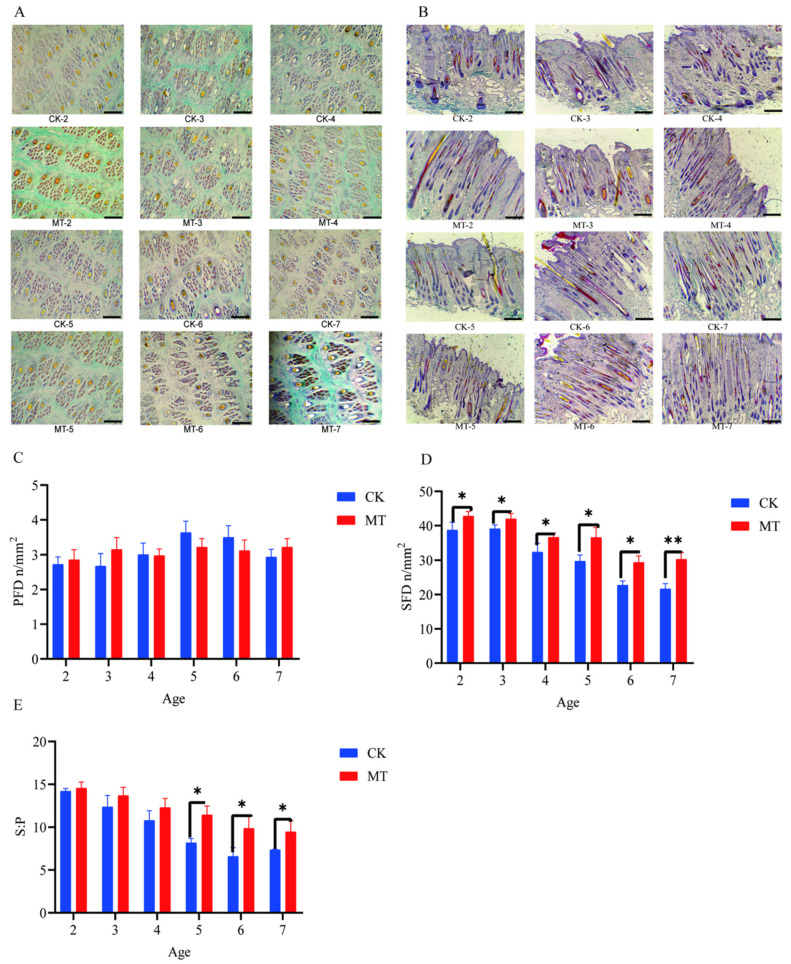
Hair follicle traits of cashmere goats treated with melatonin (MT) or not treated (CK) at different ages. (**A**) Photomicrographs of transverse sections of skin (40×, Sacpic staining); (**B**) photomicrographs of longitudinal sections of skin (40×, Sacpic staining); (**C**) primary hair follicle density (PFD); (**D**) secondary hair follicle density (SFD); (**E**) ratio secondary to primary follicles (S:P). All the values are the mean ± standard deviation. * *p* < 0.05, ** *p* < 0.01, CK vs. MT. Scale bar = 200 μm.

**Figure 3 ijms-24-03403-f003:**
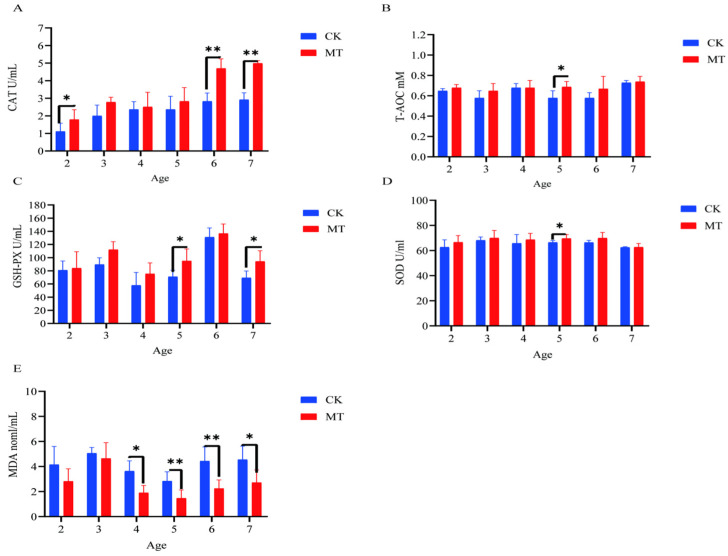
Serum antioxidant capacity of cashmere goats treated with melatonin (MT) or not treated (CK) at different ages. (**A**) Catalase (CAT); (**B**) total antioxidant capacity (T-AOC); (**C**) glutathione peroxidase (GSPH); (**D**) superoxide dismutase (SOD); (**E**) malondialdehyde (MDA). All the values are the mean ± standard deviation. * *p* < 0.05, ** *p* < 0.01, CK vs. MT.

**Figure 4 ijms-24-03403-f004:**
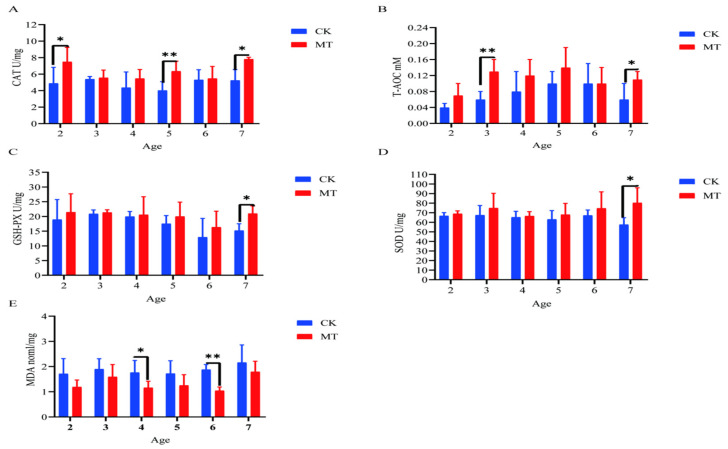
Skin antioxidant capacity of cashmere goats treated with melatonin (MT) or not treated (CK) at different ages. (**A**) Catalase (CAT); (**B**) total antioxidant capacity (T-AOC); (**C**) glutathione peroxidase (GSPH); (**D**) superoxide dismutase (SOD); (**E**) malondialdehyde (MDA). All the values are the mean ± standard deviation. * *p* < 0.05, ** *p* < 0.01, CK vs. MT.

**Figure 5 ijms-24-03403-f005:**
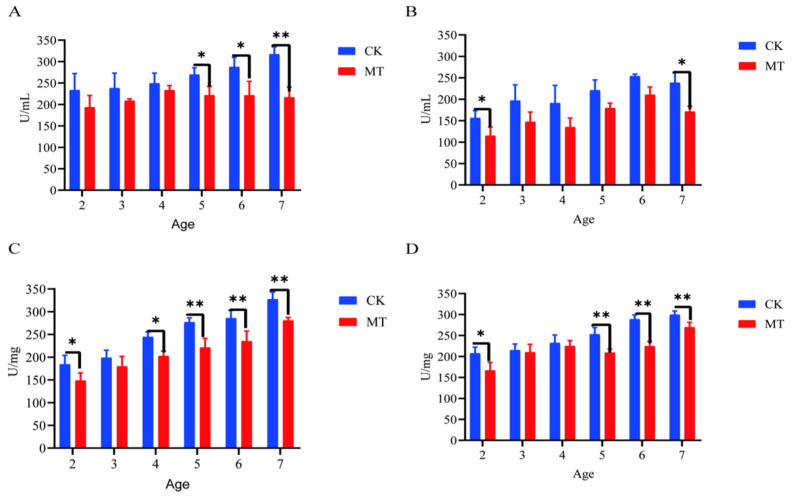
Levels of reactive oxygen and nitrogen species (ROS, RNS) in the serum and skin of Cashmere goats treated with melatonin (MT) or not treated (CK) at different ages. (**A**) ROS levels in serum; (**B**) RNS levels in serum; (**C**) ROS levels in skin; (**D**) RNS levels in skin. All the values are the mean ± standard deviation. * *p* < 0.05, ** *p* < 0.01, CK vs. MT.

**Figure 6 ijms-24-03403-f006:**
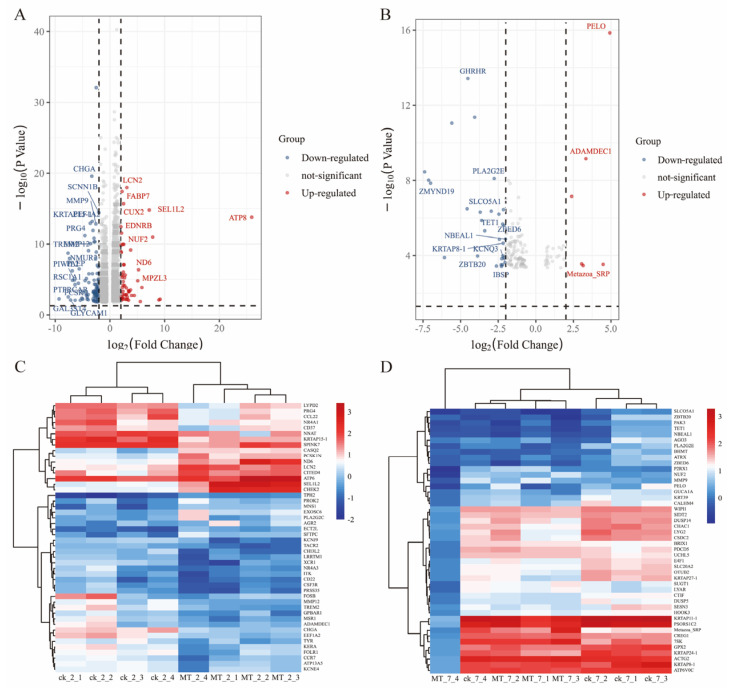
mRNA expression in the skin of cashmere goats treated with melatonin (MT) or not treated (CK). Volcano plots indicating up and downregulated mRNA transcripts in the MT group compared with the control (CK) group at (**A**) 2 and (**B**) 7 years of age. Heat maps of mRNA transcripts showing hierarchical clustering of altered mRNA transcripts in the MT and CK groups at (**C**) 2 and (**D**) 7 years of age. Up and down-regulated genes are in red and blue, respectively.

**Figure 7 ijms-24-03403-f007:**
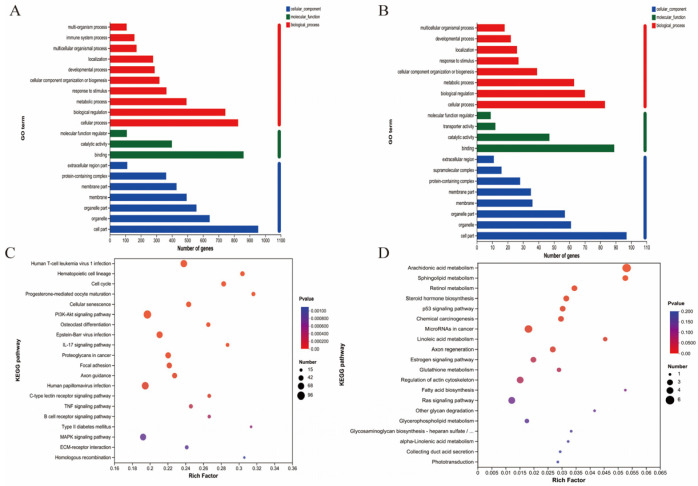
mRNA expression in the skin of cashmere goats treated with melatonin (MT) or not treated (CK) at different ages. (**A**) Gene ontology (GO) and (**C**) Kyoto Encyclopedia of Genes and Genomes (KEGG) analyses of selected differential expression genes (DEGs) in skin between MT and control at 2 years of age. (**B**) GO and (**D**) KEGG) analyses of selected DEGs in skin between MT and control at 7 years of age.

**Figure 8 ijms-24-03403-f008:**
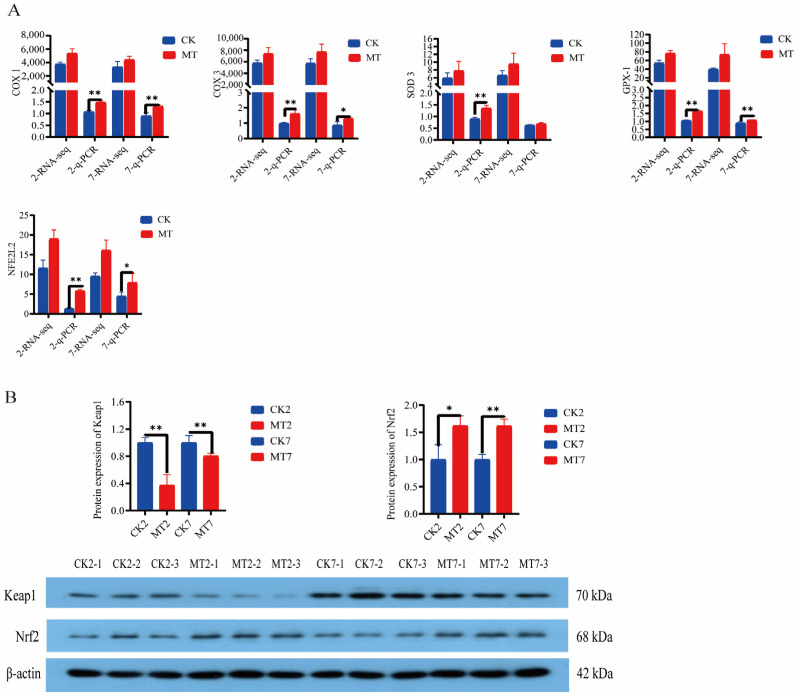
Gene and protein expression in the skin of 2-year-old and 7-year-old cashmere goats treated with melatonin (MT) or not treated (CK) at different ages. (**A**) Levels from qRT-PCR and RNAseq for: cyclooxygenase-1 (*COX-1*), cyclooxygenase-3 (*COX-3*), superoxide dismutase-3 (*SOD-3*), glutathione peroxidase-1 (*GPX-1*), nuclear factor *Nrf2*, numerals represent ages, 2 or 7 years. (**B**) Western blots showing *Keap-1, Nrf2,* and *β-actin*. All the values in A are the mean ± standard deviation. * *p* < 0.05, ** *p* < 0.01, CK vs. MT (*n* = 3).

**Figure 9 ijms-24-03403-f009:**
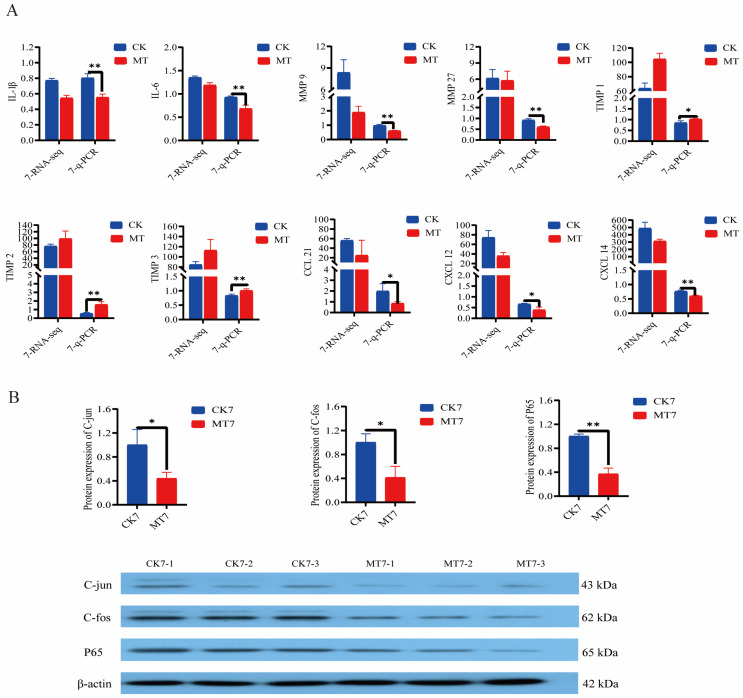
Gene and protein expression in the skin of 2-year-old and 7-year-old Cashmere goats treated with melatonin (MT) or not treated (CK) at different ages. (**A**) Levels from qRT-PCR and RNAseq for: interleukin-1β (*IL-1β*), *IL-6*, matrix metalloproteinase-9 (*MMP-9*), *MMP-27*, Tissue Inhibitor of metalloproteinase-1 (*TIMP-1*), *TIMP-2, TIMP-3*, chemokine ligand gene 21 (*CCL-21*), chemokine ligand gene-12 (*CXCL-12*), *CXCL-14*. Numerals represent ages 2 or 7 years. (**B**) Western blots showing *C-jun, C-fos, P65,* and *β-actin*. All the values are the mean ± standard deviation. * *p* < 0.05, ** *p* < 0.01, CK vs. MT (*n* = 3).

**Figure 10 ijms-24-03403-f010:**
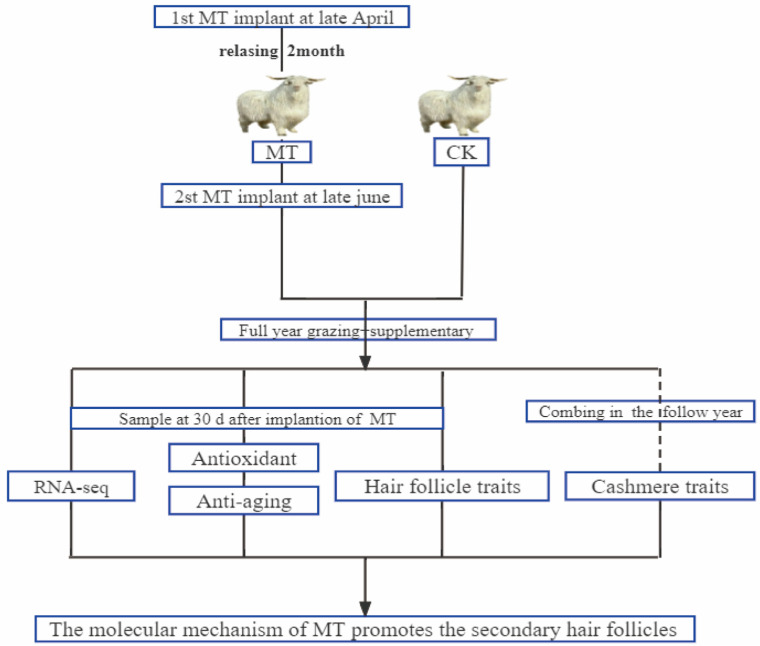
Schematic workflow of the experimental design of this study.

**Figure 11 ijms-24-03403-f011:**
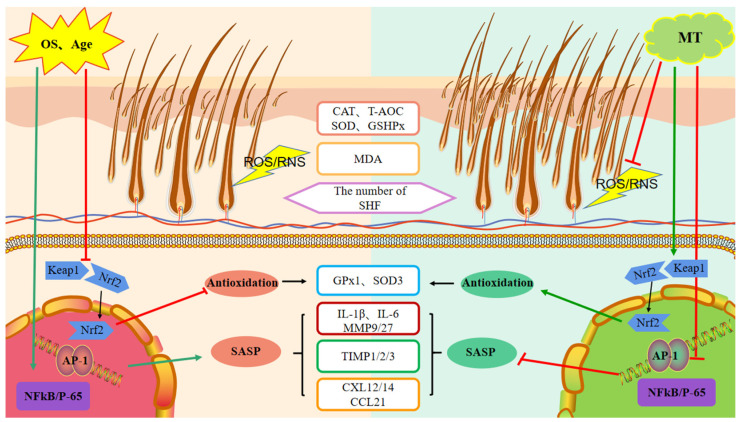
Schematic representation of the mode of action of melatonin on the promotion of secondary hair follicle growth and development. MT-melatonin, OS-oxidative stress, *CAT*-catalase, *T-AOC*-total antioxidant capacity, *SOD*-superoxide dismutase, *GSHPX/GPx*-glutathione peroxidase, *MDA*-malondialdehyde, SHF-secondary hair follicle, *Keap1*-Kelch-like ECH-associated protein 1, *Nrf2*-nuclear factor erythroid 2-related factor 2, *NFκB*-nuclear factor kappa B, SASP-senescence-associated phenotype, *IL*-interleukin, *TIMP*-tissue inhibitor of metallo proteinases, *CXL/CCL*-chemokine.

**Table 1 ijms-24-03403-t001:** The primers information of genes.

Target	Primers Sequences (5′-3′)	Product Size (bp)
GAPDH	F: ATGTTTGTGATGGGCGTGAA R: GGCGTGGACAGTGGTCATAAGT	153
MMP9	F: TAGAGAGCACGGAGATGGGTATC R: GAAGTGGGCATCTCCCTGAAT	97
MMP27	F: CTTGATGTTCCCAAACTACGTCTC R: GTAGATGGATTGGATTCCGTTG	85
TIMP1	F: GTCATCAGGGCCAAGTTCGT R: GGAAGTATCCGCAGACGCTC	169
TIMP2	F: AGGTGGACTCTGGCAACGA R: TTCAGGCTCTTCTTCTGGGTG	274
TIMP3	F: GATCAAATCCTGCTACTACCTGCC R: CCGGATGCAAGCGTAGTGTTT	121
CCL14	F: TTCAAGAGGACCTTACCACCCT R: ATGTAATGGCCCTTCTTGGTG	147
CCL21	F: CTCCAGGTCCAAGGCAGTGAT R: CAGTCATCTTTAGGGTCTGCACATA	190
SOD3	F: CACTACAACCCGATGTCCGTG R: CCGCGTTACCGTTCTCCA	214
CXCL12	F: CTACAGATGCCCTTGCCGATT R: GTTCTTCAGCCTTGCCACGAT	115
COX1	F: TTCTGATTCTTTGGACACCCTG R: AACCCGATTGATATTATGGCTC	143
COX3	F: GGAAGGAGACCGTAACCACATA R: CTACGAAGAAAGTTGAACCGTAGA	143
GPX1	F:GAAAAGTGCGAGGTCAATGGC R: ACAGCAGGGTTTCAATGTCAGG	253
IL-1β	F: TGCTGGATAGCCCATGTGTG	75
	R: TGCAGAACACCACTTCTCGG	
IL6	F:TGAAGGAAAAGATCGCAGGTCTAA R: ACCTTTGCGTTCTTTACCCACT	100
CXCL14	F:AGCATGAGGCTCCTGACCG R: TAGCGAATCTTGGGTCCCTTTC	110
NFE2L2	F:GTAGCCACTGCTGATTTAGACGA R: CCAACTTCTTTATCCAGTGAGGG	206

## Data Availability

The data used to support the findings of this study are available from the corresponding author upon request.

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
