# Peer review of "Melatonin Promotes the Development of Secondary Hair Follicles in Adult Cashmere Goats by Activating the Keap1-Nrf2 Signaling Pathway and Inhibiting the Inflammatory Transcription Factors NFκB and AP-1"

_ijms, 2023, doi:10.3390/ijms24043403_

Round 1
Reviewer 1 Report
Comments on abstract, title, and references-
- the aim of the study is very clear.
- The title is relevant to the aim of the study. This study has a good idea to study conducted to investigate the molecular mechanisms through which Melatonin exerts its effects on the development and functions of secondary hair follicles of cashmere goats at different ages.
## Comments on the introduction
This study introduces the importance of Exogenous melatonin (MT) as one of the most important promoters of the growth of secondary hair follicles and improves cashmere fiber quality. So, the topic of this paper is clear. Moreover, the research question sightly appears.
## Comments on methodology
- the methods used in this study were valid and reliable
- I am satisfied that to use of a good number of cashmere goats to collect samples.
But I have some comments on this point (its advice):
- I suggested that: characterizes the Animals, diet, and experimental design through a schematic form.
- The authors described the goats were grazed on pasture (8 h per day) and provided with supplementary feed by the feeding management system of the research facility. Water was freely available. My comment: the authors have not described the diet which contains in a table.
## Comments on data results and discussion
The paper is generally well-written and structured.
- There is no result describing the level of MT secreted by the pineal gland in response to the season in goats.
- In Lines 169, 170. Data were generally higher in the skin and secondary hair follicles of goats in the MT groups in comparison with controls (CK groups) at 2 and 7 years of age 170 (Figure 9-A). My comment here: adivce the authors to explain the mechanism of improving the antioxidant capacity and reducing markers of oxidative stress.
- Line 367 Please change the number of the figure to figure 11
- Authors describe very well the mode of action of melatonin on the promotion of secondary hair follicle growth and development in Figure 11.
## Comments on conclusions
- the conclusions answer the aims of the study.
- I think the scientists have opportunities to inform future research and more research is needed to confirm the gene functions and improve follicular activity linked to goat growth.- I suggest updating some references.
Author Response
Response to reviewer #1:
Thank you very much for reviewing our manuscript. Your comments are valuable and very helpful for revising and improving our paper. We have studied comments carefully and revised the manuscript according to your detailed suggestions. Revised portion are marked in the paper. The response are as follows:
1 I suggested that: characterizes the Animals, diet, and experimental design through a schematic form.
Response: Thanks. We added the workflow of the experimental design to the manuscript.
2 The authors described the goats were grazed on pasture (8 h per day) and provided with supplementary feed by the feeding management system of the research facility. Water was freely available. My comment: the authors have not described the diet which contains in a table.
Response: Thanks. We ensured that the MT group and the CK group were fed under the same feeding condition. The supplementary feed was purchased from the local feed company, so we did not test the nutritional composition of the feed. but we described the composition of the supplement concentrate in the ‘manuscript.4.1. Animals and management’. We hope our reply is satisfactory to you.
3 The paper is generally well-written and structured. There is no result describing the level of MT secreted by the pineal gland in response to the season in goatsï¼›
Response: Thanks. Our team previously investigated the MT level of blood of cashmere goats after MT implantation and found that the MT concentration in the MT group was significantly higher than that in the control group. Therefore, we believe that the present study is not necessary. What’s more, the determination of MT is limited by the amount of blood and skin. We will adopt your proposal in subsequent trials of melatonin application in cashmere goats. We hope my reply is satisfactory to you.
4 In Lines 169, 170. Data were generally higher in the skin and secondary hair follicles of goats in the MT groups in comparison with controls (CK groups) at 2 and 7 years of age 170 (Figure 9-A). My comment here: adivce the authors to explain the mechanism of improving the antioxidant capacity and reducing markers of oxidative stressï¼›
Response: Thanks. In the ‘Discussion’, we described the effects of MT on antioxidant indices and markers of oxidative stress respectively. According to the data of differential changes in antioxidant genes and protein (Figure 9-A), it was concluded that MT activated the Keap1-Nrf2 pathway to improve antioxidant capacity and reduce oxidative stress of secondary hair follicles as well as promoting their proliferation.
5 Line 367 Please change the number of the figure to figure 11ï¼›Authors describe very well the mode of action of melatonin on the promotion of secondary hair follicle growth and development in Figure 11.
Response: Thanks. We changed the number of the figure to figure 11.
6 I think the scientists have opportunities to inform future research and more research is needed to confirm the gene functions and improve follicular activity linked to goat growth. I suggest updating some references.
Response: Thanks. Your suggestions will guide our future trials. Currently, we are are using transcriptome technology to explore the mechanism of MT promotes the growth and development of secondary hair follicles through coding and non-coding genes. In addition, we are also planning to conduct studies on the mechanism of MT on the growth of secondary hair follicles in vitro. Finally, we modified some of the references. We hope our response satisfy you.
The above is my response. I hope my answers will satisfy you. Thank you again for your review.
Reviewer 2 Report
Please see the attachment

Author Response
Thank you very much for reviewing our manuscript. Your comments are valuable and very helpful for revising and improving our paper. We have studied comments carefully and revised the manuscript according to your detailed suggestions. Revised portion are marked in the paper. The response are as follows:
The study is supported by a comprehensive conceptual and robust analytical framework. However, there are areas requiring additional attention by the authors.
1 Abstract
The word count of the abstract is 357
The authors should report only their results and draw (concise) conclusions based on them.
Response: Thanks. We edited and revised the “Abstracts” based on your comments.
2 Results
structure of the results under the “differential mRNA expression” subheading (2.6) deviate from the other sections in that the comparison between MT and CK is initially lost while it re-emerges in the figures that follow.
Response: Thanks. We changed“differential mRNA expression” to “Differential mRNA expression between MT vs CK”.
3 Discussion
The authors should delineate results of this study from those of previously published studies. For example, known effects of MT on secondary hair follicles should not be reported as though they were results of the current study but rather should be appropriately referenced where mentioned in this discussion. Further, the study seeks to unravel the molecular basis for MT effects on the development of secondary hair follicles. Therefore, the discussion should clearly highlight the gap filled by these results and clearly contextualize them in light of similar/ related published studies.
Response: Thanks. We revised the “Discussion” in accordance with your comments.
Materials and methods
It is interesting to know how performance evaluation was conducted in this study e.g ovarian function, growth and reproduction.
Response: Thanks. We carefully considered the comments of you and other reviewers ( They think that the growth performance and reproductive performance can be deleted ), the purpose of this study was to investigate the mechanism by which MT affects hair follicle growth and cashmere traits, growth performance and reproduction did not seem to be very important. Therefore, we also considered it more appropriate to delete parts of the manuscript concerning growth performance and reproductive performance. We hope our response satisfy you.
Conclusions
Authors should firm up key conclusions for example targeted application of MT (>5 years of age based on their findings) and elucidation of its mode of action. The authors are silent about possible confounder/ interaction effects.
Response: Thanks. In conclusion, we pointed out that MT is more effective in promoting cashmere in older goats. In addition, given that there are too many age groups, and the effect of interaction may not be that significant. So,we chose not to analyze the interaction effects of MT. However, I think your suggestions are very constructive, and we will consider your suggestions in the future study of MT application in the goat. We hope our response satisfy you.
References
A total of 82 publications are appropriately included in the list of references in this paper.
Response: Thanks. We modified some of the references.
Abstract
Lines 23-24 : how were the effects of MT on growth and reproductive performance evaluated?
Response: Thanks. After considering the suggestions of three reviewers, we deleted the content about the growth performance and reproduction performance.
Lines 25-28: should be rephrased into one concise sentence
Response: Thanks. We made modifications according to your suggestion.
Line 31: did the authors mean “significant effects on genes” or “differences in expression of genes”?
Response: Thanks. We modified it as “ differences in expression of genes...”.
Lines 31-34: should be rephrased clearly indicating that the comparison is between the treatment and control groups and a distinction should be made between the gene and the protein it codes for.
Response: Thanks. In the previous results, we compared the gene and protein expression of MT and CK, and highlighted the role of MT through two pathways in the conclusion. We have distinguished the coding genes and proteins as you suggested.
Lines 34-36: This conclusion needs to be firmed up. It is important to note that adult goats were aged between 2 and 7 years and therefore, the conclusion should highlight in which age group(s) the MT effects were significant.
Response: Thanks, As you suggested, we changed it to“Collectively, these effects of exogenous MT enhanced the quality and yield of cashmere fibers,especially in 5-7 years old.”
Line 37: did the authors mean older and not elderly?
Response: Thanks, We modify it to ‘older’.
Line 38: how were MT effects on skin aging assessed?
Response: Thanks, We believe that MT can delay skin aging from two points: 1) the number of hair follicles and the number of hair follicle group; the number of hair follicles in young skin are more than those in aging skin. In addition, under the microscope, it can be intuitively observed that the development of hair follicles in MT group is better than that in control group. 2) The secretion of SASP in aging skin increases, and the expression of SASP gene in MT group is different from that in control group. Therefore, we believe that MT can delay the aging of skin. We hope our response satisfy you.
Introduction
Lines 53-54: there seems to be a contradiction between these two lines
Response: Thanks, We revised this two lines.
Line 66: at what age are cashmere goats considered adults?
Response: Thanks, 1 years old is considered an adult.
Line 72: is it a decline in hair follicles or a decline in development of hair follicles or decline from death (apoptosis) of hair follicles? This sentence should be rephrased to communicate what the authors intend to state.
Response: Thanks, We modify it as “It is believed that factors such as oxidative stress, accumulation of reactive oxygen and nitrogen species (ROS, RNS), skin aging, and deterioration of the microenvironment of the hair follicles are responsible for the decline in the number of hair follicles[21,22], and the secretion of MT also declines with age[23,24].
Line 73: is it predicted or hypothesized? If predicted, how was the prediction made?
Response: Thanks, We modify it as “So, we hypothesized that...”
Line 78: it is not clear why the authors are interested in the various age groups (2-7 years) while their study focuses on adult cashmere goats. This goes back to the comment made at line 66.
Response: Thanks.The reason why we chose cashmere goats between 2 and 7 years old is related to secondary hair follicle development. Hair follicles are divided into the primary hair follicles and the secondary hair follicles. The primary hair follicles grow hair, the secondary hair follicles grow cashmere. The hair follicles occur in the embryo, the primary hair follicles occur before the secondary hair follicles. The secondary hair follicles are derived from the dermis around the primary hair follicles. After the goat kids enter the first year of life, secondary hair follicles begin to undergo apoptosis and the cashmere begin to fall off. While the old secondary hair follicles die, the new secondary hair follicles begin to grow and enter the next cycle. In most cashmere goat research study, 2-3 years old is chosen because the body condition of cashmere goats at 2-3 years old is better than that of goat kids and older goats in hair follicle development and cashmere characteristics. In the cashmere production process, the cashmere quality decreases with the increase of cashmere goats age, and in cashmere goat pasture, the ratio of 4-7 years old accounts for 70% of the all cashmere goat age. Furthermore, based on previous research results, melatonin can improve the antioxidant capacity of young kids, so as to promote the growth and development of secondary hair follicles. We believe that MT may increase the number of secondary hair follicles by anti-oxidation in young and prime cashmere goats, and may also increase the number of secondary hair follicles by anti-aging in old goats. in conclusion, we selected cashmere goats aged 2-7 years for this study. We hope our response satisfy you.
Results
Line 82: is it “no effects” or “no statistically significant effects”? It is important for the methodology to highlight how MT effects on live weight were assessed. It is difficult to understand the herd structure without pedigree and other herd level information included in the methodology.
Lines 84-87: legend and caption should communicate the same thing. It is in unclear as to what CK-4 means?
Line 88: What were the baseline values and how did these change within groups?
Response: Thanks. We deleted the content of “2.1Growth performance and breeding performance”.
Line 134: Labels on the y-axes should include marker names/ codes as done in Figures 4 and 5 for easy reading
Response: Thanks. We explained it in the annotation of the figure. We hope our response satisfy you.
Line 138: It is not clear why results for only 2 and 7 year old age groups are presented here. The rationale seems to be that the two groups have extremely divergent values for the parameters under study. However, this being an exploratory study, including comparison across the groups would contribute valuable information.
Response: Thank you for your suggestion. As you said, we chose 2 and 7 years old as subjects for the molecular mechanism because we took into account cashmere traits, hair follicle traits and blood-related indicators; Secondly, it was found that genes related to secondary hair follicles were significantly different between 2 and 7 years old in the RNA-seq data in our previous studies. However, we think your suggestion is very constructive and we will consider your suggestion in future experiments. We hope our response satisfy you.
Lines 154-159: Is it possible to improve the legibility of the figures?
Response: The format for figures is TIFF. The legibility of figures does not change depending on the size of the figures.
Line 189: add a space between metalloproteinases and (TIMP-1 and Line 250: add “in” between expressed and skin
Response: Thanks. We made modifications according to your suggestion.
Lines 251-252: check the use of singular and plural forms (its vs their)
Response: Thanks. We unified the singular forms according to your suggestion.
Line 260: hair loss and alopecia mean the same thing
Response: Thanks. We change it as “...hair aging or alopecia[65]”.
Materials and methods
Line 291: Adults 2-7 years of superfine cashmere goats were used in he study. It is not clear if these were clones. Further, the sampling frame is not clear- did each age group have 15 MT and 15 CK goats? What were the selection criteria in addition to age? Live body weight, performance history (parturition, birth weight, weaning weight, yearling weight, Average Daily Gain, Sex?
Response: Thanks. Cashmere goats in this study were all pure Inner Mongolia Albas white cashmere goats with the same pedigree background. We selected 180 female cashmere goats with similar live weight and good body condition in each age, 15 in control group and 15 in MT group. We hope our response satisfy you.
Line 297: What was the feeding management system? This is important given that malnutrition affects both reproduction and growth as well as leads to oxidative stress.
Response: Thanks. Feeding management system is that the production of cashmere goats under grazing, supplementary feeding and free drinking water, which mainly includes basic maintenance growth, cashmere growth and reproduction. Our experiment was carried out under the condition of ensuring that the MT group and the control group were raised in the same way, and meeting the production requirements of cashmere goats. We hope our response satisfy you.
Line 318: name of the software used?
Response: Thanks. We change it as ‘The software ( DESeq2 and Goatools ).....’
Line 325: were the primers designed by the authors or adapted from a published article?
Response: Thanks. The primers were designed by we team.
Line 352: Were interaction terms significant? Particularly those involving treatment, sex, age group?
Response: Thanks. In this study, the effect of MT on promoting cashmere in different ages was mainly discussed, and the interaction of MT, gender and age was not taken into account, but the possibility of interaction cannot be denied. We will consider your suggestion in the later study.We hope our response satisfy you.
Line 355: “<” already implies all less than
Response: Thanks. We deleted “or below”
Line 367: The term AGE is not defined in the caption
Response: Thanks. We made changes to the Figure11.
Line 380: Ethics approval applies to all on-station/ laboratory or on-farm trials/ studies involving animals. However, country specific requirements are duly noted.
Response: Thanks. The ethical declaration was attached to “The ethical declaration should be attached”
The above is my response. I hope my answers will satisfy you. Thank you again for your review.
Reviewer 3 Report
First of all, I congratulate the authors on their work and wish them every success. The study is generally well designed and the results are well presented. Apart from a few minor corrections that I have mentioned below, the study is very valuable.
· Further details and studies on Inner Mongolia cashmere goats and the genetics of cashmer traits can be added to introduction.
· If the growth traits are to be kept in the manuscript, further information should be given on the growth and reproductive traits in the introduction section.
· The study was designed on secondary follicular development and presented appropriately. In this context, the effect of melatonin on growth and reproductive properties can be excluded from the manuscript, since the effect of melatonin is outside the objective of the study.
· Discussion part is weak and can be discussed with further studies. In addition to the cashmer quality of goats, it can also be discussed with the studies of fine wool sheep hair follicle development.
· The ethical declaration should be attached to the manuscript.
Author Response
Thank you very much for reviewing our manuscript. Your comments are valuable and very helpful for revising and improving our paper. We have studied comments carefully and revised the manuscript according to your detailed suggestions. Revised portion are marked in the paper. The response are as follows:
1 Further details and studies on Inner Mongolia cashmere goats and the genetics of cashmere traits can be added to introduction.
Response: Thanks. We added the content about further details and studies on Inner Mongolia cashmere goats and the genetics of cashmere traits to introduction.
2 If the growth traits are to be kept in the manuscript, further information should be given on the growth and reproductive traits in the introduction section.
Response: Thanks. At your suggestion, We deleted the section on growth performance.
3 The study was designed on secondary follicular development and presented appropriately. In this context, the effect of melatonin on growth and reproductive properties can be excluded from the manuscript, since the effect of melatonin is outside the objective of the study.
Response: Thanks. We think your advice is very good. The purpose of the article becomes clearer and the structure is more reasonable after removing growth performance and reproductive performance.
4 Discussion part is weak and can be discussed with further studies. In addition to the cashmere quality of goats, it can also be discussed with the studies of fine wool sheep hair follicle development.
Response: Thanks. According to your requirements, we added the content of cashmere and secondary hair follicles in the ‘Discussion’.
5 The ethical declaration should be attached to the manuscript.
Response: Thanks. The ethical declaration was attached to “The ethical declaration should be attached”.
The above is my response. I hope my answers will satisfy you. Thank you again for your review.